# Uncertainty for Proximal Femur Fractures Classification

**Mayar Lotfy**\*  MAYAR.MOSTAFA@TUM.DE

*Chair of Computer Aided Medical Procedures, Technical University of Munich, Germany*

**Selina Frenner**\*  SELINA.FRENNER@GMAIL.COM

**Marc Beirer**  MARC.BEIRER@SBK-VS.DE

**Peter Biberthaler**  PETER.BIBERTHALER@TUM.DE

*TUM School of Medicine, Technical University of Munich, Germany*

**Shadi Albarqouni**  SHADI.ALBARQOUNI@UKBONN.DE

*Clinic for Diagnostic and Interventional Radiology, University Hospital Bonn, Germany*

*Helmholtz AI, Helmholtz Munich, Germany*

## Abstract

Deep Learning methods over the past years provided high-performance solutions for medical applications. Yet, robustness and quality control are still required for clinical applicability. In this work, the uncertainty of proximal femur fracture classification was modeled. We introduce a reliability measure to our predictive model using the Monte Carlo Dropout approach. We performed an extensive quantitative and qualitative analysis to validate the results. We further exposed the results to expert physicians in order to get feedback on the model's performance and uncertainty measures. Results demonstrate a positive correlation between the misclassification of the model's prediction and high uncertainty scores. Additionally, the uncertainty measures are mimicking the actual radiologist's uncertainty for challenging examples reflected on intra- and inter- experts variability.

**Keywords:** Deep Learning, Uncertainty, Quality Control, Radiology, Proximal femur fractures.

## 1. Introduction

In the realm of medical diagnosis, Deep Learning has emerged as a powerful tool for pattern recognition, showcasing remarkable advancements in recent years. Studies have shown that integrating automated AI models with emergency medicine clinicians significantly enhances their ability to accurately detect fractures, ultimately improving patient outcomes (Lindsey et al., 2018). Among the most prevalent fractures globally are proximal femur fractures, where prompt diagnosis and treatment are crucial for patient well-being and even survival (Schroeder et al., 2022). However, accurate diagnosis heavily relies on the experience of medical professionals (Plant et al., 2015). To further enhance diagnostic accuracy, computer-aided diagnosis (CAD) systems hold immense potential in reducing errors, optimizing treatment costs, and saving time in future medical practices (Gale et al., 2017). Nevertheless, to ensure the successful integration of such systems into clinical routines, evaluating not only the overall performance but also the reliability of individual diagnoses is essential. This study aims to address this need by introducing Monte Carlo dropout (MCDO) layers (Kendall and Gal, 2017), a state-of-the-art approximation of Bayesian Neural Networks, as a quality control measure. The incorporation of these layers provides a reliable uncertainty score for automated proximal femur fracture classification, following the AO-fracture classification system.

---

\* Contributed equally

## 2. Methodology

Having a dataset consisting of 1347 X-ray images and their corresponding labels $y \in C$, where $C$ represents one of three subsets in different classification scenarios, namely $C \in \{C_1, C_2, C_3\}$; $C_1 \subset \{Fracture, Normal\}$ for fracture detection, $C_2 \subset \{A, B, Normal\}$ for the three-class scenario, and $C_3 \subset \{A1, A2, A3, B1, B2, B3\}$ for the six-class scenario, our objective is to develop a model for femur fracture classification that assigns each image a class label along with uncertainty scores. The database used in this study was collected from 672 patients by the trauma surgery department of Klinikum Rechts der Isar. The ground truth labels were collected by three different experts, and all images were verified by a senior radiologist. Additional annotations were obtained from three independent experts for the testing set, with each expert providing two independent readings conducted at different times, considering variations in shifts and lighting conditions. To assess the uncertainty scores qualitatively, an independent radiologist re-evaluated 30% of the test set.

## 3. Experiments and Results

The focus of our work is not to outperform the SOTA, but rather to capture the uncertainty while achieving comparable performance. The experiments were designed in a way to analyze the performance of MCDO under 1) Stochastic (with MCDO) vs. Deterministic networks, and 2) different loss functions; namely Cross Entropy (CE), Weighted Cross Entropy (WCE), and Focal Loss (FL) (Lin et al., 2017), and 3) different network architectures.

**Implementation.** ResNet model was adopted from (Jiménez-Sánchez et al., 2018), where the MCDO was introduced only at the last dense layer, treating the rest as a deterministic network. As for the stochastic DenseNet model, the dropout layers were introduced at each convolutional layer and in the transition blocks, where the same hyper-parameters were adopted from (Huang et al., 2017). Besides, 5-fold cross-validation was conducted for the DenseNet models. To evaluate the performance of our model, we compute the confusion matrix, *F1*-score and the macro average *F1*-score for each classification scenario to account for class imbalance.

**Results.** In general, models introduced with MCDO layers achieved comparable F1-score performance to their own baseline models. In an attempt to analyze and compare the performance of MCDO with different settings, namely, deterministic (ResNet) and Stochastic (DenseNet) models against the individual readings of the three experts and the majority consensus, we visualize the Receiver operating characteristic (ROC) curves in Fig.2. Overall, the performance of our stochastic DenseNet models performed similarly compared to the expert's readings. Further, the coherence between the uncertainty scores and misclassification was qualitatively measured. Results demonstrated that the misclassified images mostly occur in the highly uncertain region. This is most apparent in cases of fracture detection and 3 classes of classification. As for 6 classes classification, it shows a Gaussian distribution for the uncertainty scores with almost no coherence with misclassification. This scenario in particular is the most challenging out of the previous cases and is aligned with the F1-score reported. This confirms two key outcomes; First, the uncertainty score is a reliable measure for detecting mistakes in the model performance and a valid robustness quality control. Second, the model's performance is reflected by how well and coherent the

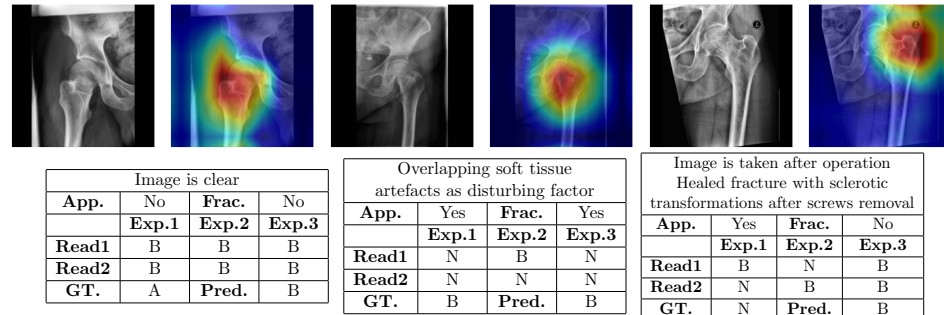

| | Image is clear | | |
|---|---|---|---|
| App. | No | Frac. | No |
| | Exp.1 | Exp.2 | Exp.3 |
| Read1 | B | B | B |
| Read2 | B | B | B |
| GT. | A | Pred. | B |

| Overlapping soft tissue artefacts as disturbing factor | | | |
|---|---|---|---|
| App. | Yes | Frac. | Yes |
| | Exp.1 | Exp.2 | Exp.3 |
| Read1 | N | B | N |
| Read2 | N | N | N |
| GT. | B | Pred. | B |

| Image is taken after operation Healed fracture with sclerotic transformations after screws removal | | | |
|---|---|---|---|
| App. | Yes | Frac. | No |
| | Exp.1 | Exp.2 | Exp.3 |
| Read1 | B | N | B |
| Read2 | N | B | B |
| GT. | N | Pred. | B |

Figure 1: **Qualitative Assessment:** In the 3-class scenario, the assessment reveals the following (from left to right): Low Uncertainty Misclassified, High Uncertainty Correctly Classified, and High Uncertainty Misclassified cases.

modeling of uncertainty, i.e. ResNet+ vs. DesneNet+ (*cf.* Fig.1). Lastly, the coherence between the calculated uncertainty scores of the test images and the uncertainty of radiologists' annotation was validated. In a like manner, the aforementioned scores with the inter- and intra-observer variability of the three independent experts who participated in providing two distinctive reads each were compared. The radiologist was asked to provide comments on the image and the fracture specifying if the classification is an easy or challenging task, along with stating if the difficulty comes from the fracture complexity (i.e. cognitive) or from the appearance (i.e. perceptual). To this end, we expect a reflection of the high uncertainty on the appearance and the fracture difficulty, also a reflection on the disagreement and the inter-/intra- variability among the three experts.

## Conclusion

This paper aimed to compare the performance of various networks in classifying proximal femur fractures while incorporating an uncertainty score for quality control. The results demonstrated a strong correlation between misclassification and uncertainty scores, with the DenseNet stochastic implementation exhibiting the highest alignment with misclassi-

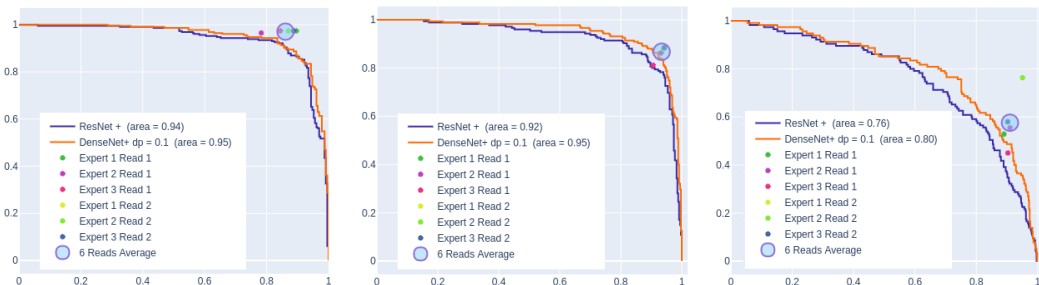

Figure 2: **Clinical experts vs. Our model.** Comparison of different architecture models and the clinical experts for the 2, 3 and 6 classes respectively from left to right.

fied cases. This implies that a high uncertainty score indicates a higher risk of prediction errors. Moreover, through qualitative analysis, it was observed that the uncertainty measures closely mirrored the actual uncertainty experienced by radiologists when dealing with challenging and complex cases, as evident from intra- and inter-expert variability. These findings have important implications both in scientific research and clinical applications. In research, they can be utilized to enhance the training of computer-aided diagnosis (CAD) systems by identifying errors and addressing difficulties, particularly in complex classifications. Additionally, they can serve as a crucial element in facilitating the clinical implementation of deep learning models by providing clinicians with a quantitative measure of quality for CAD predictions. Future work should focus on enhancing the robustness of the models and expanding the analysis to different datasets, including other anatomical regions of the human body.

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
