# OpenReview forum: "Uncertainty for Proximal Femur Fractures Classification"
_MIDL.io/2023/Short_Paper_Track — MIDL 2023 Short paper track Poster_

### Official Review · Reviewer_vdmX · 2023-04-23
**Uncertainty measure agrees with radiologists assessment**

**Rating:** 7
**Confidence:** 3

**Review:**

The paper describes an approach for uncertainty measurement using Monte Carlo dropout, here applied to proximal fractures classification.

Pros:
* Classification scenarios with hierarchical classes helps understanding
* Data has multiple annotators
* Evaluation with radiologists

Cons:
* Figures very hard to read due to size
* Quantitative results with radiologists not clear, is there a measure of inter-observer agreement?

---

### Official Review · Reviewer_QZk8 · 2023-04-25
**Review for: Uncertainty for Proximal Femur Fractures Classification**

**Rating:** 5
**Confidence:** 4

**Review:**

The authors present an uncertainty analysis for prediction methods for femur fractures, using the dropout bootstrap method in two types of feed-forward convolutional neural networks.

While the topic of this work is very interesting, I think the experimental section leaves multiple results under-reported or unreported. We should see a quantitative assessment of the correlation between misclassification and the measured uncertainty; instead statements to this effect only appear in the text, and without numerical values.

The plots in Figure 2 are tiny. Space constraints are understood, but surely these plots are more important than the saliency examples of Figure 1? We should also see error-bars or other uncertainty representation in the plots of Figure 2, or some other plot (matrix of predicted vs actual of uncertainty density estimates?) showing how these estimates interact with the predictions, since this correlation is reported.

The results in the text are interesting: classification error correlating with neural network uncertainty says to me that something interesting is happening in the networks at those particular images, and the fact that this trend appears to carry over between network architectures AND experts is even more promising. I encourage the authors to dig into these results and continue onward with this project.

However, the reporting of results in the submitted work leaves much to be desired.

Minor notes:

miss-classification -> misclassification